# A Predictive Study of Resilience and Its Relationship with Academic and Work Dimensions during the COVID-19 Pandemic

**DOI:** 10.3390/jcm9103258

**Published:** 2020-10-12

**Authors:** Silvia San Román-Mata, Félix Zurita-Ortega, Pilar Puertas-Molero, Georgian Badicu, Gabriel González-Valero

**Affiliations:** 1Nursing Department, University of Granada (Spain), Campus Universitario de Melilla, Calle Santander 1, 52005 Melilla, Spain; silviasanroman@ugr.es; 2Department of Didactics of Musical, Plastic and Corporal Expression, University of Granada (Spain), Campus de Cartuja, s/n, 18071 Granada, Spain; felixzo@ugr.es (F.Z.-O.); javiconde@correo.ugr.es (P.P.-M.); ggvalero@ugr.es (G.G.-V.); 3Department of Physical Education and Special Motricity, Faculty of Physical Education and Mountain Sports, Transilvania University of Brașov, 500068 Brașov, Romania

**Keywords:** resilience, emergency services, COVID-19, adversity, lockdown

## Abstract

Background: The aim of the present study was to describe the resilience levels in a Spanish population during the Coronavirus (COVID-19) pandemic and to analyze the existing associations between high resilience and socio-demographic, work, and academic parameters. Method: 1176 individuals aged 18–67 years participated in a descriptive cross-sectional study. The participants were administered the 10-item resilience scale developed by Connor-Davidson (CD-RISC-10) and an ad-hoc questionnaire that collected information on socio-demographic, work, and academic variables. Basic descriptive data were used to statistically analyze the data, and a binary logistic regression model was developed incorporating the professional occupation, academic level, whether the respondent worked in emergency services, and whether the respondent had dependents. Results: Slightly more than a quarter of the participants showed low resilience, almost half reported moderate resilience, and slightly more than a quarter had high resilience. Those who were employed were 2.16-times more likely to have high resilience, whilst those with higher education were 1.57-times more likely. Those working in emergency services were 1.66-times more likely, and those with dependents were 1.58-times more likely to have high resilience. Conclusion: In addition to the relationships found, a need to improve the resilience levels in the population was found.

## 1. Introduction

Currently, we find ourselves with a serious public health problem and in a state of global emergency due to the Coronavirus (COVID-19) pandemic [1,2]. This virus first appeared in China at the end of 2019 and in just a few weeks had spread to many other countries around the globe [3,4]. It was responsible for a high number of deaths in the first months after its emergence [5]. Spain was in confinement during the first two weeks of the state of alarm. According to the latest data provided by the Ministry of Health [6], on the 15 March, the number of cases reported nationwide amounted to 7753 (2000 new cases in the last 24 h), representing 16.52 cases per 100,000 inhabitants, including 288 deceased and 382 admitted to the intensive care units.

In this sense, citizens around the globe and those governing them were pushed to take quick and efficient action, complying with the instructions given by the relevant health authorities to slow down this pandemic [7]. The COVID-19 pandemic appears to be a risk factor for psychological illness and sleep disorders, reportedly having a significant impact on these constructs [8,9]. Studies, such as the one carried out by Forte, Favieri, Tambelli, and Casagrande [10] revealed that individuals reported levels of general psychopathological symptoms, anxiety, and post-traumatic stress disorder (PTSD) symptoms that were higher than the cut-off scores.

Resilience capacity, understood as a set of intrinsic factors that characterizes all individuals and is implicated in the process of overcoming adversity [11,12], takes on great importance in relation to the degree of success that can be achieved by health measures. Resilience capacity is also crucial for the promotion of psychological wellbeing in the population [13,14].

Thus, resilience makes up one of the most important dynamic psychological factors with regard to the protection and adaptability of individuals, being malleable to development and improvement through intervention programs [15,16]. In fact, a large amount of scientific literature, currently available within various populations, deals with resilience in general terms, whether this be with children [17,18], adolescents [19], or other populations [20].

In the same way, this skill is currently a hot topic of study in health workers and those working in emergency services because of the arduous work they perform and the stress it entails [21,22,23,24,25]. Studies carried out with patients with different pathologies at different stages of the disease process are also highly relevant, especially those conducted with oncology patients [26,27,28]. In this way, patients who present lower resilience show greater problems in relation to emotional regulation and, as a consequence, higher levels of stress and anxiety. Such findings were stated by Vaughan et al. [29]

Likewise, research is found that relates resilience with socio-demographic, religious [30], personal, and family factors, in addition to academic performance [31]. It must also be stated that various opinions exist with respect to the resilience capacity and its relationship with gender [32], with studies also emerging within transgender groups [33,34].

Thus, the importance of the resilience capacity to human behavior is clear. As a result of this, confinement and the limitation of movement as a result of the global state of health emergency in which Spain is embroiled, presents a type of adversity to be overcome by the whole population. In this way, there is a need to identify the levels of resilience in the population during this time of crisis. Based on the scarcity of studies tackling the topic of COVID-19 and its relationship with the way in which adversity is tackled and overcome, the present study was proposed with the following objectives: (1) describe the levels of resilience in the Spanish population during the pandemic, and (2) analyze the existing associations between high resilience and socio-demographic, work, and academic parameters.

## 2. Method

### 2.1. Participants and Procedure

A total of 1176 individuals from Spain participated in the present descriptive and cross-sectional research study. The participants were aged between 18 and 67 years (M = 35.35 years; SD = 11.900), with 457 (38.9%) being male and 719 (61.1%) being female. The sample was selected through a process of random sampling. In order to be selected, individuals had to be in full possession of their psychological faculties, provide informed consent, be of adult age, not be retired, and not suffer from any type of condition that would impede participation in the research. These requirements made up the inclusion and exclusion criteria. The sample was obtained from all Spanish cities, requesting participation from all those who agreed to do so voluntarily. The study sample was collected in two periods depending on the different states of lockdown. The first period was from 15 to 22 March (*n* = 727; 61.8%) and the second period was from 23 to 31 March (*n* = 449; 38.2%).Two questions from the self-reported survey were duplicated to avoid bias in the responses and to check that they were not filled in randomly. We excluded 171 questionnaires after detecting that they had been incorrectly filled out or that the recipients did not meet the inclusion criteria. 

The participants were contacted through various calls placed on social networks in diverse social groups so that the sample would be as random as possible. Once contact was made, the potential participants were informed regarding how to fill out the document, informing them that all collected data would be kept totally anonymous and used only for research purposes. The researchers were present in a virtual way during data collection in order to guarantee correct implementation of the process and resolve any doubts. This was achieved by providing a personal Google Meets link associated with the group of researchers, to enable users to connect with the researchers and resolve any questions they might have. The present research received approval from the ethics committee of the University of Granada (641/CEIH/2018).

### 2.2. Variables and Instruments

The self-registration form (ad-hoc questionnaire), collected data in relation to sex (male or female), age, whether the respondents were responsible for dependents during confinement (older individuals or relatives), whether the respondents knew somebody in their environment who had suffered from or had COVID-19, occupation prior to confinement (student, neither working nor studying, state employee, works with the public, self-employed, or works for a private company), the highest academic level achieved (basic studies, professional training, higher studies (up to baccalaureate), postgraduate studies, or doctorate studies), whether they work in emergency services (categorized as yes or no), and the period of study completion structured according to period 1 (from the 15–22 March) and period 2 (from the 23–31 March).

Resilience test. The Spanish version of the 10-item resilience scale developed by Connor-Davidson (CD-RISC) was used. This comes from the original version of the CD-RISC proposed by Connor and Davidson [35] and adapted into Spanish by Notario-Pacheco et al. [36] and Soler-Sánchez, Meseguer-de Pedro, and García-Izquierdo [37]. The scale is formed by 10 items which request respondents to provide ratings along a Likert type scale thatruns from 0 (totally disagree) to 4 (totally agree). Questions include the example: “I am capable of adapting to changes”. Initial studies obtained Cronbach alpha values higher than 0.80, with α = 0.87 reported by Soler-Sánchez et al. [37], and α = 0.85 reported by Notario-Pacheco et al. [36], whilst the present study obtained a value of α = 0.89.

### 2.3. Data Analysis

For the basic descriptive analysis and cross-tabs, the statistical software package IBM SPSS^®^ was used in version 25.0 for Windows. The normality and homogeneity of the sample was established through the Kolmogorov–Smirnov test.

Following this, binary logistic regression (odds ratio and 95% confidence intervals) was performed. High resilience provided the exposure variable as it was one of the specific objectives being considered. The model examined its association with socio-demographic, work, and academic variables. Likewise, Cox and Snell’s R^2^ was employed to examine the model fit, whilst the Hosmer–Lemeshow test was used to determine the goodness of fit. Variables were introduced into the model manually if they met the criteria of having shown significant associations in prior bivariate analysis carried out through crosstabs. Variables that did not show significance at this prior stage were excluded from the model (*p* ≥ 0.05).

Given that the proposed analysis was binary, participants with low and moderate resilience were categorised into a single “not high resilience” group. Likewise, given the dichotomous nature of analysis, the remaining variables were categorised as follows: Sex (0 = female and 1 = male), professional occupation (0 = employed and 1 = not employed), level of study (0 = without higher education and 1 = higher education), professional occupation related with the emergency services (0 = not emergency services and 1 = emergency services), responsible for dependents (0 = does not have dependents and 1 = has dependents), associated with individuals with COVID-19 (0 = not associated with anybody with COVID-19 and 1 = associated with somebody with COVID-19), and time-period (0 = period from the 15–22 March and 1 = period from the 23–31 March).

## 3. Results

The 1176 participants of the present study were distributed between both genders, with 61.1% being female and 38.9% being male. Of these, 66.3% stated being responsible for dependents (older adults or children), whilst 33.7% did not. A total of 27.3% were categorized as having high resilience, 46.2% had moderate resilience, and 26.5% had low resilience. With regard to the question asking whether individuals in their immediate environment had contracted COVID-19, 42.7% indicated yes, whilst 57.3% stated that no. In reference to respondents’ professional occupation prior to confinement, 34% were public employees, 22.3% were self-employed or lent their services to a private company, 21.8% were students, 18.5% reported studying and working, and 3.4% were neither studying nor working. With regard to their academic level, 50.1% reported their highest level of study being “studies of higher education”, 29.3% had postgraduate qualifications, 11.1% had professional training, 6% possessed basic studies, and 3.7% had only third grade studies (Table 1)

Finally, with regard to working in emergency services, 27.9% reported doing so relative to 72.1% who did not. In relation to the time-period as it relates to the state of alarm, 61.8% of the questionnaires were completed during period 1 (from the 15–22 March) and 38.2% during period 2 (from the 23–31 March) (Table 1)

In the relational study of variables relating to the resilience level, statistically significant differences were found (*p* = 0.015) pertaining to sex. Specifically, low resilience was more common amongst females than males (29.1% relative to 22.5%), with these figures being inverted when high resilience was considered (31.3% for males and 24.9% for females). With regard to being responsible for dependents, differences were shown in the data (*p* = 0.000), with individuals responsible for dependents showing a greater prevalence of high resilience (35.1%) than those without this responsibility (23.3%).

No association was found (*p* = 0.248) with regard to whether respondents had individuals in their immediate environment who had contracted COVID-19, whilst participants’ occupation prior to confinement did produce statistically significant differences (*p* = 0.001). Concretely, participants who were working as public employees, were self-employed, or worked for a private company obtained higher values, with 27.5% reporting a high resilience relative to just 16.4% of students who reported the same optimum level.

Regarding the academic levels, statistically significant differences (*p* = 0.001) emerged. In this case, 33% of respondents with postgraduate or doctorate studies reported high resilience, this being a greater percentage than the 18.6% of individuals with only basic studies who also obtained scores belonging to this category. In relation to whether or not individuals worked in a position related to emergency services, a statically significant association was found (*p* = 0.002). This was generated because those who did have a relevant profession (32%) presented a higher prevalence of high resilience than those who did not come into contact with emergency services through their work (25.5%). These results were inverted when considering low resilience, with 29.2% of those in contact with emergency services falling into this category, relative to 19.5% of those not in contact. Finally, differences were not detected (*p* = 0.243) with regard to the period of study completion and resilience level. (Table 2)

Once the descriptive and relational study was determined, we proceeded to the second study objective, which was to establish a predictive model of high resilience through binary logistic regression. The variables describing close others with COVID-19 and the time-period (*p* ≥ 0.05) were excluded. In the first step of analysis, sex did not produce significant outcomes and so it was also excluded from the model. In the second step, good fit was shown through outcomes of the omnibus test (X^2^ = 48.721; 4df; sig = 0.000), Hosmer–Lemeshow test (X^2^ = 4.095; 6df; sig = 0.664), Cox and Snell R^2^ (0.041), and Nagelkerke statistic (0.059). The model adequately explained 72.7% of cases. 

Likewise, as can be seen in the following table, the model identified associations (*p* < 0.05 in the adjusted regression model) between resilience and professional occupation (Exp [B]: 2.160 [1.504–3.101]), academic level (Exp [B]: 1.579 [1.089–2.290]), job related to emergency services (Exp [B]: 1.668 [1.242–2.239]), and responsibility for dependents (Exp [B]: 1.583 [1.194–2.097]) (Table 3).

## 4. Discussion

The present research work was conducted with a sample of 1176 Spanish adults, with 61.1% being represented by females and 38.9% by males. The study sought to determine associative patterns between various aspects associated with resilience, and socio-demographic, work, and academic aspects during the period of confinement caused by the global COVID-19 crisis. In the opinion of the authors of this manuscript, no studies with the characteristics of the present study have been developed to date. Although the results should be considered with caution, positive and predictive relationships of resilience capacity were established with the variables of this study. This study can be used as a starting point for developing resilience-based interventions to help minimize psychological consequences during the COVID-19 pandemic.

First, one of the descriptive results produced in relation to the resilience capacity was that almost half of participants reported values that corresponded to moderate resilience. This finding is similar to that reported by Rodríguez and Ortunio [38], who specified highly similar percentages for this dimension. On the other hand, Szu-Ying et al. [39] stated that slightly more than half of older adults with cardiovascular problems who made up the study sample reported low resilience. This could be due to the fact that they have an illness that causes them stress. 

Along similar lines, it is appropriate to highlight that four out of every ten participants in the present study reported having had contact with individuals affected by COVID-19. This could provide an explanation for the medium resilience levels. Fear of the unknown coupled with uncertainty regarding future socio-economics and health can generate mental health problems in the population including the consumption of toxins, somatization, stress, anxiety, and depression that can lead to the risk of suicide [40,41,42]. 

When observing scores pertaining to “low resilience”, females predominated, whilst males were mostly found in the group of those with “high resilience”. This coincides with other studies [12,22,32,38] that allude to both cultural and traditional factors. In this sense, males were traditionally considered responsible for the economic wellbeing of their family. This brought with it skill acquisition and decision making based on the need to take care of their loved ones, leading to rises in resilience. 

At the same time, entirely contrasting positions were also found, such as that reported by Laul et al. [43]. These authors showed females to be more resilient, whilst further studies indicated that females were more resilient due to their role as mediator and overseer of the family, with the education and care of members being one of their main functions [44]. Other studies failed to note any sex-based differences, although they did establish resilience-based relationships within health workers [45]. 

On the other hand, it is interesting to point out the emergence of studies carried out in recent years, especially in transgender groups [33,34]. In the present day, the topic of resilience and gender is somewhat controversial. In this study, sex had to be excluded from the predictive model in the first step of model construction as it was not found to be significantly related. 

In relation to professional occupation, the results of the present study indicated that public employees, self-employed workers, and those working at private companies had higher resilience scores. This fact could be explained by the management and planning skills and competencies of individuals at a cognitive, planning, and intellectual level. Individuals draw on these resources to perform tasks, whether this be in the work, social, or academic setting. These resources act as protective factors and help develop resilience [46]. In addition, Zhang et al. [47] added the factor of job commitment as another potentially influential factor.

Thus, at the beginning of this year, a study was conducted in Korea targeting workers that included health workers, in which high levels of work stress, anxiety, and depressive feelings were associated with suicidal ideation, and they found that there was an inverse relationship between resilience and suicidal ideas; that is, high levels of resistance in professionals decreased the incidence of self-injurious thoughts [48]. 

In fact, Mckinley et al. [45] argued in their study that better scores were obtained by hospital doctors relative to others working in medicine in general. This is likely because dedication and job commitment are greater in hospitals, and this could be considered to be one of the factors related with increased resilience. In the present study, similar figures were found in the proposed regression model, with the likelihood of having high resilience being 2.16-times greater amongst those who were regularly employed. 

On the other hand, the scarcity or lack of material necessary to carry out the work with the necessary safety harasses the work environment, favoring the discomfort of the health workers at the mental level. In this sense, the lack of prevention in health supplies was linked to the risk of contagion, user demand, patient morbidity, and the risk of death increasing the burden of stress on health personnel, as well as increasing pressure, concern, and anxiety levels in said personnel [49]. In this line, García-Fernández et al. [50] found, in their research, higher levels of stress and anxiety in health professionals who considered the level of protection in their work environment insufficient or inadequate, while those who had access to adequate protection presented emotional well-being.

At the same time, high resilience scores were uncovered in staff working in the context of emergency services. The ability to overcome adverse situations, adapt to them, and come out stronger the other end, was greater in individuals with stressful occupations as they require emotional management skills to carry out their work [21,22,24,25]. Studies have not only been performed directed towards this population; in fact, research has also been developed at a community level in relation to resilience in hospitals, given the importance of resilience when delivering quality care to service users [23]. In particular, resilience in this setting acts as a protective factor for emergency staff against the potential psychological malaise they might suffer when performing their responsibilities. Thus, our regression model indicated that professionals dedicated to emergency medicine were 1.66-times more likely to have high resilience. 

Along similar lines, Zhang et al. [47] outlined the implications of working in different occupational professions as another individual factor. This may influence resilience whilst at the same time improve the quality of work performance, in addition to having a positive association with spiritual health and self-concept [51,52].

All of these observations are also related with the existence of intervention programs designed to improve the resilience levels in nursing staff. An example of such a program is the one developed by Henshall, Davey, and Jackson [53]. This identified the importance of this skill when practicing in the health profession and demonstrated the achievement of better scores following university training [54]. Similarly, in other care settings, such as in social work, resilience is integrated into educational programs, courses, and professional development frameworks. This is due to its importance in carrying out these activities, as has been indicated by Clevelant, Warhurst, and Legood [55]. 

In this way, academic level is considered to be another factor associated with resilience. Individuals who possess higher education qualifications were shown to be more resilient. This collaborates previously conducted work by Orkaizaguirre-Gómara et al. [31], which found that resilience in university students increased as they progressed through each academic year. Likewise, the results presented by Szu-Ying et al. [39] found higher resilience levels amongst individuals with higher education qualifications. For this reason, this variable was included in the developed regression model, with the outcome confirming that students undertaking higher education were 1.57-times more likely to show high resilience. 

Considering the factors we previously mentioned in relation to careers, it is fitting to highlight that individuals with children or dependents were also found to be more resilient than those without. This is illustrated in the regression model via an association between both variables. Thus, it appears that, in some way, the responsibility to care for others leads to development of this skill in such a way that participants with dependents were 1.58-times more likely to present with high resilience. 

Finally, with regard to the limitations presented by the present work, there is a scarcity of existing literature conducted on resilience within Spanish populations during the COVID-19 pandemic. Research in other contexts was found in relation to this topic. With regard to the transgender population, no relevant items were included within the gender variable, which could be considered as another study limitation. Another limitation could be the homogeneity of the sample, as it was mostly women who were more willing to answer the survey. Thus, no distinction was made between age groups, this being a possibility for future studies.

Finally, given that Spain found itself in a state of national emergency and health crisis at the time of data collection, it would have been interesting to include the variable of “stress”. This would be likely to provide additional information given the large number of studies that have related high resilience with good stress management [56,57] and fewer depressive symptoms [58]. Studies, such as that conducted by Moksnes and Lazarewicz [19], found resilience to play a compensatory role in the relationship between stress and emotional symptoms.

## 5. Conclusions

In conclusion, individuals who were employed (2.16 times), had higher education (1.57 times), worked in the emergency services setting (1.66 times), and were responsible for dependents (1.58 times) were more likely to report high resilience. In addition to these associations, a clear need was demonstrated to improve resilience levels in general. For this reason, intervention programs must be conducted to develop this capacity from early ages, whilst its dynamic and malleable nature also makes targeting the general population worthwhile. In the same way, future research directives should include additional variables, such as stress, anxiety, and depression.

Uncovering further variables related to the COVID-19 pandemic will also provide priceless information regarding how to manage the stress caused. Similarly, it will be useful to develop psychosocial studies and implement intervention programs that are adapted toward the development of this capacity, whilst also promoting psychological well-being in the population.

## Figures and Tables

**Table 1 jcm-09-03258-t001:** Descriptive characteristics of the sample.

**Sex**	Male	38.9% (*n* = 457)	Has dependents	Yes	33.7% (*n* = 396)
	Female	61.1% (*n* = 719)		No	66.3% (*n* = 780)
**Resilience**	Low	26.5% (*n* = 312)	Close other with corona virus (COVID)-19	Yes	42.7% (*n* = 502)
	Medium	46.2% (*n* = 543)		No	57.3% (*n* = 674)
	High	27.3% (*n* = 321)	Age	M = 35.35; DT = 11.900	
**Occupation**	Full-time student	21.8% (*n* = 256)	Academic level	Professional training	11.1% (*n* = 130)
	Public employee	34% (*n* = 400)		Doctorate	50.1% (*n* = 589)
	Neither studies nor works	3.5% (*n* = 41)		Postgraduate	29.3% (*n* = 344)
	Studies and works	18.5% (*n* = 217)		Third grade studies	3.7% (*n* = 43)
	Self-employed/works at a private company	22.3% (*n* = 262)		Basic education	6% (*n* = 70)
**Works in the Emergency Services**	Yes	27.9% (*n* = 328)	Time-point	Period 1	61.8% (*n* = 727)
	No	72.1% (*n* = 848)		Period 2	38.2% (*n* = 449)

**Table 2 jcm-09-03258-t002:** Associations between resilience and all other variables.

Variables		Resilience			
		Low (*n* = 312)	Medium (*n* = 543)	High (*n* = 321)	*Sig*
Sex	Male	22.5% (*n* = 103)	46.4% (*n* = 212)	31.3% (*n* = 142)	0.015 *
	Female	29.1% (*n* = 209)	46.0% (*n* = 331)	24.9% (*n* = 179)	
Has dependents	Yes	22.7% (*n* = 90)	42.2% (*n* = 167)	35.1% (*n* = 139)	0.000 *
	No	28.5% (*n* = 222)	48.2% (*n* = 376)	23.3% (*n* = 182)	
Close others with COVID-19	Yes	24.5% (*n* = 123)	48.8% (*n* = 245)	26.7% (*n* = 134)	0.248
	No	28.0% (*n* = 189)	44.2% (*n* = 298)	27.7% (*n* = 187)	
Occupation	Full-time student	33.6% (*n* = 86)	50% (*n* = 128)	16.4% (*n* = 42)	0.001 *
	Public employee	26.3% (*n* = 105)	46% (*n* = 184)	27.8% (*n* = 111)	
	Neither studies nor works	24.4% (*n* = 10)	53.7% (*n* = 22)	22% (*n* = 9)	
	Studies and works	25.3% (*n* = 55)	41.9% (*n* = 91)	32.7% (*n* = 71)	
	Self-employed/works at a private company	21.4% (*n* = 56)	45% (*n* = 118)	33.6% (*n* = 88)	
Academic level	Professional training	26.2% (*n* = 34)	50.0% (*n* = 65)	23.8% (*n* = 31)	0.001 *
	Doctorate	27.7% (*n* = 163)	48.6% (*n* = 286)	23.8% (*n* = 140)	
	Postgraduate	24.4% (*n* = 84)	41.9% (*n* = 144)	33.7% (*n* = 116)	
	Third grade	27.9% (*n* = 12)	23.3% (*n* = 10)	48.8% (*n* = 21)	
	Basic education	27.1% (*n* = 19)	54.3% (*n* = 38)	18.6% (*n* = 13)	
Works in emergency services	Yes	19.5% (*n* = 64)	48.5% (*n* = 159)	32.0% (*n* = 105)	0.002 *
	No	29.2% (*n* = 248)	45.3% (*n* = 384)	25.5% (*n* = 216)	
Time-point	Period 1	25.0% (*n* = 182)	46.4% (*n* = 337)	28.6% (*n* = 208)	0.243
	Period 2	19.0% (*n* = 130)	45.9% (*n* = 206)	25.2% (*n* = 113)	

Note 1. Statistically significant differences at the level *p* < 0.05 *.

**Table 3 jcm-09-03258-t003:** Binary logistic regression model.

	B	Standard Error	Wald	df	Sig.	Exp(B)	95% CI for EXP(B)
Lower	Upper
Professional occupation	0.770	0.185	17.415	1	0.000	2.160	1.504	3.101
Academic level	0.457	0.189	5.819	1	0.016	1.579	1.089	2.290
Emergency services	0.511	0.150	11.579	1	0.001	1.668	1.242	2.239
Dependents	0.459	0.144	10.227	1	0.001	1.583	1.194	2.097
Constant	−2.282	0.247	85.259	1	0.000	0.102

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
