# Peer review of "A Predictive Study of Resilience and Its Relationship with Academic and Work Dimensions during the COVID-19 Pandemic"

_jcm, 2020, doi:10.3390/jcm9103258_

Round 1

Reviewer 1 Report

This study aims to describe resilience levels in a Spanish population during the COVID-19 pandemic and to analyze existing associations between high resilience and socio-demographic, work, and educational variables. Information on socio-demographic, employment, and educational variables was collected from 1,176 adults. Participants also completed the CD-RISC-10 resilience scale. Results showed that slightly more than a quarter of participants showed low resilience, almost half reported moderate resilience, and slightly more than a quarter had high resilience. Resilience was higher in people employed, with higher education, in those working in emergency services and that had dependents.

The study is very interesting, but some revisions are needed. Some suggestions follow.

1) In the abstract, give the full name of CD-RISC-10.

2) In the abstract, participants were defined as older adults. However, the method reported: "Participants were aged between 18 and 67 years". Please clarify this point.

3) What do the authors refer to with "ad hoc test"? If they refer to the collection of socio-demographic data, these are collected through interviews or questionnaires, which are not tests. Please clarify this point.

4) The study was performed during two periods: period 1 (from the 15th to the 22nd of March) and 38.2% 145 during period 2 (from the 23rd to the 31st of March). Please report the number of respondents for these two periods in the Method section.

5) It would be useful to introduce a table with data relating to infections, deaths, and the availability of places in intensive care in the two periods in Spain. It would also be helpful to report the government measures taken during these two periods. Finally, it would be useful to have information on the availability of beds in hospitals and medical personnel during the periods considered. I believe that such data can easily be found on Spanish institutional sites.

6) It could be useful to expand the literature on the impact of the COVID-19 pandemic on psychological well-being and distress in the general population. Resilience is associated with other psychological and physical conditions (e.g., anxiety, depression, sleep quality, etc.). In this regard, I suggest reading these articles, in which other references can be found: 

  • Casagrande, M., Favieri, F., Tambelli, R., & Forte, G. (2020). The enemy who sealed the world: Effects quarantine due to the COVID-19 on sleep quality, anxiety, and psychological distress in the Italian population. Sleep Medicine.
  • Favieri, F., Forte, G., Tambelli, R., & Casagrande, M. (2020). The Italians in the time of Coronavirus: Psychosocial aspects of unexpected COVID-19 pandemic. Frontiers. Available at SSRN 3576804." 
  • Forte, G., Favieri, F., Tambelli, R., & Casagrande, M. (2020). The Enemy Which Sealed the World: Effects of COVID-19 Diffusion on the Psychological State of the Italian Population. Journal of Clinical Medicine, 9(6), 1802.
  • Forte, G., Favieri, F., Tambelli, R., & Casagrande, M. (2020). COVID-19 Pandemic in the Italian Population: Validation of a Post-Traumatic Stress Disorder Questionnaire and Prevalence of PTSD Symptomatology. International Journal of Environmental Research and Public Health17(11), 4151.

7) Some limitations of the study should be reported. For example, the participants were mostly women. Did being young, middle-aged, or elderly influence the results? 

8) The discussion should be enriched, discussing the results according to the consequences that the COVID-19 pandemic, and the following restrictive measures adopted to counter the spread of the virus, has produced on psychological and behavioral levels.

9) I suggest highlighting better the novelty of this study in the field of the COVID-19 pandemic.

10) A linguistic revision would be made.

Author Response

Dear Reviewer, 

I attach the comments. 

Thank you!

Reviewer 2 Report

The topic of the article is very useful and necessary in the times that the whole planet lives, my congratulations for choosing this topic. However there are several issues that I think need to be improved:

Introduction:

-In the first place I think lines 64-66 would belong to the Conclusions section rather than to the Introduction.

Method:

- In lines 81-83 they indicate "It must be indicated that 171 questionnaires were excluded after detecting that they had been incorrectly filled out or that recipients did not meet inclusion criteria, but in lines 87-89 they say:" Researchers were present in a virtual way during data collection in order to guarantee correct implementation of the process and resolve any doubts ".
How then 171 questionnaires were completed incorrectly?

- In lines 85-87, how were the participants contacted for the correct completion of the questionnaire and how were each of the surveys developed?

In summary, although the subject of the article is appropriate to the context and the current problem, the methodology used does not provide control over the sample, the inclusion criteria are vague to consider them as such (anyone could fill it in without having a high probability that the data is correct, meets the criteria, or is not duplicated). It does not include an independent variable that can verify the object of study. In short, it does not guarantee the credibility of the data.

Good luck with the article

Author Response

(The authors gave the same response as above.)

Round 2

Reviewer 2 Report

The authors have not made the suggested change regarding the location of an introductory paragraph, where they clearly make proposals that should be included in the Conclusions section on the development of psychosocial studies and to implement intervention programs adapted to the development of this capacity. promoting at the same time the psychological well-being of the population.

The authors have not justified the use of the methodology used and how they have ensured the accuracy of the data. The use of an adequate methodology will determine the veracity of the rest of the article. The use of social networks for the collection of quantitative data may not be the best way to represent the whole of a country, since the scope of data collection is not limited and there is no control over it.

Author Response

Dear Reviewer, 

I attach the completions. 

Thank you!
